# Effects of Different Processing Methods on the Quality and Physicochemical Characteristics of *Laminaria japonica*

**DOI:** 10.3390/foods12081619

**Published:** 2023-04-12

**Authors:** Zuomiao Yang, Xueting Li, Meiqi Yu, Shan Jiang, Hang Qi

**Affiliations:** 1National Engineering Research Center of Seafood, Liaoning Provincial Aquatic Products Deep Processing, Technology Research Center, School of Food Science and Technology, Dalian Polytechnic University, Dalian 116034, China; 2Haide College, Ocean University of China, Qingdao 266003, China

**Keywords:** *Laminaria japonica*, cooking methods, processing properties, physicochemical properties, flavor, structure

## Abstract

The effects of four domestic cooking methods, including blanching, steaming, boiling, and baking treatments, on processing properties, bioactive compound, pigments, flavor components, and tissue structure of *Laminaria japonica* were investigated. The results showed that the color and structure of kelp changed most obviously after baking; steaming was most beneficial in reducing the color change of the kelp (Δ*E* < 1), while boiling was most effective in maintaining the texture of the kelp (its hardness and chewiness were close to that of raw kelp); eight volatile compounds were detected in raw kelp, four and six compounds were detected in blanched and boiled kelp, while eleven and thirty kinds of compounds were detected in steamed and baked kelp, respectively. In addition, the contents of phloroglucinol and fucoxanthin in kelp after the four processing methods were significantly reduced (*p* < 0.05). However, of all the methods, steaming and boiling were the best at preserving these two bioactive substances (phloroglucinol and fucoxanthin) in kelp. Therefore, steaming and boiling seemed more appropriate to maintain the original quality of the kelp. Generally, to improve the sensory characteristics of each meal of *Laminaria japonica* and to maximize the retention of active nutrients, several different processing methods are provided according to the respective effects.

## 1. Introduction

As consumers around the world become more cautious about nutritious and healthy foods and their ingredients, the concept of nutrition is changing rapidly. Recently, seaweed has gradually become an important ingredient in contemporary recipes due to its different forms, textures, flavors, and nutrients as well as its rich nutrient content of bioactive compounds. *Laminaria japonica* (*L. japonica*), also known as kombu and kelp, belongs to a kind of brown algae. It is generally 2~4 m long, 20~30 cm wide, and in the shape of a flat ribbon. The *L. japonica* growing in seawater is also a marine vegetable with dual edible and medicinal values. In addition, *L. japonica* contains significant varieties of compounds such as protein, amino acids, functional polysaccharides, dietary fiber, and some trace elements [1,2]. Moreover, it is also a good source of polyphenol compounds and pigments such as phlorotannins, fucoxanthin, and chlorophyll [3]. Clinical and experimental studies also have found that the bioactive components extracted from *L. japonica* have anti-tumor, anti-coagulation, blood-pressure-lowering, blood-sugar-lowering, blood-lipids-regulating [4], anti-viral, anti-oxidation, anti-mutation, and immune-enhancing functions [5]. At present, there are many studies on the cultivation technology, nutritional value, and medicinal value of *L. japonica* [6,7], but there are relatively few studies on *L. japonica* processing as an edible food for human consumption.

In recent years, food processing has gradually become a research focus. During the cooking treatment of vegetables, taste quality plays a crucial role in consumers’ acceptance and consumption of food [8]. People have conducted a vast amount of research on various cooking methods and their effects [9]. However, due to different processing conditions, the quality of the final product also shows varying degrees of change. Nwosisi et al. [10] evaluated the effects of different thermal treatments (baking, pressure cooking, and open cooking) on the texture of six cultivars of sweet potato. The results indicated that the different cooking methods affected the textural parameters prominently. Compared with the rest of the processing methods, hardness, resilience, chewiness, and gumminess of pressure-cooked sweet potato were significantly reduced. Under baking conditions, gumminess, chewiness, and springiness were the highest. Danowska-Oziewicz et al. [11] analyzed color parameters and sensory properties of the broccoli and green asparagus cooked by different methods. The results show that all cooking treatments caused intense green color, visible taste, and crunchy but soft texture compared with the both raw samples. Moreover, when studying the influence of processing methods (boiling, steaming, microwaving, and pressure cooking) on quality of Galega kale, it was found that steaming kept the kale’s nutritional contents extremely well while also giving it the worst sensory quality [12]. Hence, studying the changes in the quality of *L*. *japonica* under different cooking methods will help improve the sensory characteristics of each meal and determine the ideal changes in the main processing characteristics and the best cooking methods required.

Different processing methods also cause different changes in the bioactive components, pigments, flavor components, and structure of vegetables. Thermal treatments (such as boiling, steaming, microwaving, baking, and frying) cause changes in the phenolic content of components. It was indicated that steaming had a significant effect on maintaining the polyphenols of sweet potato leaves [13]. Dos Reis et al. [14] evaluated the extraction of carotenoids, flavonoids, phenolic compounds, and chlorophyll from fresh inflorescences of broccoli and cauliflower under different cooking conditions (boiling, steaming, microwaving, and sous vide) compared to the concentrations measured in fresh vegetables. Because of the short cooking time and high temperature, the content of bioactive substances in the cooked broccoli was higher than that resulting from other cooking methods. The total phenol content and pigment of artichokes, green beans, cauliflower, and carrots were studied under different cooking methods (boiling, sous-vide cooking, and water-immersion cooking). The preservation degree of phenolic content, carotenoid, and chlorophyll of all the tested vegetables by sous-vide cooking was higher than that by boiling [15]. Sánchez et al. [16] observed the impact on the content of carotenoids and chlorophyll of six kinds of vegetables under high-pressure (HP) and high-pressure and high-temperature (HPHT) treatments. In terms of chlorophyll, there was no increase or degradation in HP treatment at 20 °C. HPHT treatment (70 °C and 117 °C) degraded both types of chlorophyll, but chlorophyll a was less stable at 70 °C than chlorophyll b. One possible reason is that chlorophyll a is converted to pheophytin more easily than chlorophyll b or that chlorophyll b is more stable during thermal treatment than chlorophyll a [17].

Jiang Shan et al. [18] studied the effects of traditional cooking methods (blanching, steaming, boiling, and baking) on the color, texture, and bioactive nutrient content of *Undaria pinnatifida* (*U. pinnatifida*), and the results showed that blanching and boiling were more beneficial than other cooking methods in maintaining the quality of *U. pinnatifida*. In addition, Jiang Shan et al. [19] further investigated the effects of domestic cooking methods (air frying (AF), microwave, and high-temperature and -pressure (HTP) cooking) on the quality, nutrient content, and bioactive substances of *U. pinnatifida*, and the results showed that of the three processing methods, microwave cooking had a more favorable effect in maintaining the quality of *U. pinnatifida*. Akomea-Frempong et al. [20] reported on the effects of blanching, freezing, and fermentation on the physicochemical, microbiological, and organoleptic quality of sugar kelp (*Saccharina latissima*). The results showed that the simplest cooking processes, such as blanching and freezing, could preserve the quality of the kelp and extend its short shelf life. Lafeuille et al. [21] studied the effect of temperature and cooking time on the physicochemical properties and sensory potential of aqueous extracts of seaweed from *Palmaria palmata* and *Saccharina longicruris*. They found that the aqueous extract preserved the nutrients in the crude seaweed and that the effect of temperature and cooking time depended on the composition and the seaweed. Cooking methods and other processing methods might have an effect on seaweed quality, but studies on the differences in processing characteristics and bioactive components of *L. japonica* are incomplete. Domestic cooking methods such as blanching, steaming, boiling, and baking are the most common methods of seaweed processing and are suitable for home cooking and are preferred by consumers. Therefore, in order to fill this research gap, the aim of this study was to assess the effects of four domestic cooking methods (blanching, steaming, boiling, and baking) on the processing characteristics, phloroglucinol, pigments, structure, and flavor component of the *L. japonica*. In addition, we also observed and analyzed the microstructural changes of kelp treated by different processing methods through SEM to reveal the effect of cooking methods on the texture of kelp from a microscopic perspective. In this study, the best cooking method to maximize the retention of active substances and to improve the processing characteristics of the product was investigated.

## 2. Materials and Methods

### 2.1. Materials and Reagents

The broad-leaf dried *L. japonica* brand Wang Yijia was purchased from Metro Supermarket in Dalian, China, and the place of production was Fengxian District, Shanghai, China. Phloroglucinol (≥99.0% (HPLC)) was purchased from Dr. Mao; fucoxanthin and chlorophyll a were purchased from Sigma-Aldrich Co. (Shanghai, China). All standard reference materials, gallic acid, and Folin–Ciocalteu’s phenol reagent were purchased from Sigma-Aldrich Co. (Shanghai, China). Methanol and acetone were HPLC grade, and other reagents were analytical-grade reagents produced by local companies in China.

### 2.2. Preparation of Sample

The dried *L. japonica* was rehydrated for 1 h with deionized water, its roots and top leaf tips were removed, and its middle parts were saved and were cleaned. After rehydration, *L. japonica* samples were cut into 3 cm × 3 cm pieces. The products were divided into 100 g aliquots and submitted to a subsequent heat treatment.

### 2.3. Cooking Cnditions

The pre-processed *L. japonica* samples were cooked in four different cooking methods (blanching, steaming, boiling, and oven baking). The cooking conditions were determined through preliminary experiments. The *L. japonica* samples were divided into 100 g portions and cooked by the following cooking methods:

Blanched: Raw *L. japonica* samples were added to 1 L boiling, unsalted tap water under normal pressure and removed from the water after 5 min.

Steamed: *L. japonica* samples were placed in a steamer and cooked for 30 min at 100 °C.

Boiled: Raw *L. japonica* samples were added to 1 L boiling, unsalted tap water under normal pressure and removed from the water after 30 min.

Baked: Raw *L. japonica* samples were placed in the oven and baked for 15 min; heating mode: 160 °C, hot air, heat up, and down.

At the end of each cooking test, one part was directly removed from the cooking vessel and cooled with cold water for rapid cooling. Immediately after cooling, the products were analyzed for moisture, color, texture, and aroma characteristics. The other part was placed in an oven at 50 °C for 16 h. The dried samples were crushed with a homogenizer and passed through a 200-mesh sieve. The crushed products were placed into a vacuum bag, vacuumed, and sealed at 4 °C for storage. Furthermore, the different cooking experiments were carried out in triplicate on different days to account for the changes in ingredients.

### 2.4. Processing Characteristics

#### 2.4.1. Color Analysis

The color of *L. japonica* was measured using a UltraScan Pro chromameter (UltraScan Pro, HunterLab, Reston, VA, USA); the instrument was calibrated using a white and a black tile. The total color difference (Δ*E*) was calculated using *L** (lightness) and color coordinate (*a**, *b**) values using the formula described by Armesto et al. [12] previously. In the CIE color system, in this case, *L** values indicate the intensity of color (a higher value means a lighter color), while the negative *a** color coordinate and the positive *b** color coordinate describe the intensity of green and the yellow colors, respectively.
(1)$$\Delta E_{\mathrm{ab}}=\sqrt{{\left(L_{2}*-L_{1}*\right)}^{2}+{\left(a_{2}*-a_{1}*\right)}^{2}+{\left(b_{2}*-b_{1}*\right)}^{2}}$$
where *L*_1_*, *a*_1_*, and *b*_1_* are raw values. *L*_2_*, *a*_2_*, and *b*_2_* are the values of the sample subjected to different thermal treatments. Measurements were taken at three points per replicate. Altogether, color readings were recorded from nine replicate samples per treatment [22].

#### 2.4.2. Texture Profile Analysis (TPA) Measurement

For TPA, the *L. japonica* sample was analyzed by a TA. XT. plus texture analyzer equipped with a flat plunger 5 mm in diameter (P/5) (SMS, London, UK). The parameters were set as follows: compression variable: 30%; the speed of the test: 1 mm/s; and the height of the plunger: 15 mm. TPA of *L. japonica* processed by each method was measured nine times.

#### 2.4.3. Moisture Measurement

The water state of raw and cooked samples was determined using a MesoQMR23-060H L-NMR (Suzhou (Shanghai) Niumag Electronic Technology Co., Ltd., Shanghai, China). Niumai software can be applied to measure and analyze the relaxation time of L-NMR. Approximately 0.1 g of sample was put in the L-NMR apparatus. The spine spin relaxation (T2) was measured based on the CPMG sequence. The parameter settings were as follows: P1 (22 MHZ) and P2 (44 MHz). The moisture distribution of the kelp processed by each method was measured six times.

### 2.5. Determination of Polyphenolic Compounds Content

Polyphenolic compounds were extracted with only minor modifications as described by Ummat et al. [23]. Briefly, both raw and cooked dried *L. japonica* samples were mixed with 50% ethanol (1:15, *w*/*v*) and ultrasonicated for 30 min. Then, the mixture was incubated in a THZ-82 constant temperature oscillator (Zhiborui Instrument Manufacturing Co., Ltd., Changzhou, China) at 50 °C under 120 rpm for 7.5 h. Supernatant was collected and filtered with Whatman #1 filter paper (Whatman International Limited, Maidstone, UK). The supernatant was then stored in a sealed, light-proof bottle and stored at 4 °C until use. Extract was determined in triplicate.

Preparation of phloroglucinol standard curve: Firstly, 1 mg of phloroglucinol standard was dissolved in 1 mL of chromatographic methanol, and the phloroglucinol standard stock solution was prepared at a concentration of 1 mg/mL and stored at −20 °C at low temperature and protected from light. Then, a certain amount of phloroglucinol stock solution was diluted to 0.05, 0.1, 0.5, 1.0, 5.0, 10.0, and 50.0 μg/mL of phloroglucinol standard working solution, and 1 mL was aspirated into a brown injection bottle for measurement. The equation of the standard curve was obtained as y = 653.64x + 28,389.5, where y is the peak area, and x is the concentration of phloroglucinol, mg/mL; the correlation coefficient R^2^ = 0.9972.

Determination of the phloroglucinol of kelp: Firstly, 5 mL of *L. japonica* polyphenol crude extract was rotary steamed at 40 °C to obtain a dry extract; then, 5 mL of chromatographic methanol was added to the dry extract. The obtained solution was filtered through 0.22 μm nylon microporous filter membrane, and then, 1 mL was transferred to a brown sample bottle until further use. Next, phloroglucinol was determined at 40 °C using high-performance liquid chromatography (HPLC, Shimadzu-SPD-20A, Shimadzu (Shanghai) Experimental Equipment Co., LTD, Tokyo, Japan) on a C18 column (4.6 × 250 mm, 5 µm), with some modifications according to Wang et al. [24] and Bai et al. [25]. The chromatographic separation was carried out with a mobile phase of methanol and ultrapure water at a flow rate of 0.8 mL-min^−1^, an elution gradient of 100 ~ 85% methanol, an elution time of 10 min, and a sample volume of 10 μL for detection at 280 nm by a diode array detector (DAD). The corresponding peak areas of the untreated and processed kelp samples were then determined according to the method described above, and the content of phloroglucinol in the kelp extract was calculated by substituting into the standard curve. The phloroglucinol content of kelp processed by each method was measured three times.

### 2.6. Determination of Fucoxanthin Content

Raw and cooked dried *L. japonica* samples were mixed with 80% ethanol (1:10, *w*/*v*) and sonicated for 30 min. The resulting mixture was centrifuged at 4000 rpm for 10 min. The supernatant was then evaporated (200 rpm, 50 °C) in a rotary evaporator (SY-2000, Shanghai Yarong Biochemical instrument Factory, Shanghai, China). Then, an equal proportion of chromatographic methanol was used to reconstitute. The obtained solution was filtered through 0.22 μm nylon microporous filter membrane, and then, 1 mL was transferred to a brown sample bottle until further use.

Preparation of standard curves for fucoxanthin: Firstly, 1 mg of fucoxanthin standard was dissolved in 1 mL of chromatographic methanol to prepare a stock solution of 1 mg/mL of algal fucoxanthin, which was stored at −20 °C at low temperature and protected from light. The stock solution was then diluted to 0.05, 0.1, 0.5, 1.0, 5.0, 10.0, and 50.0 μg/mL and placed in a brown injection vial for measurement. The standard curve equation was obtained as y = 223,619x − 42,866, where y is the peak area, and x is the concentration of fucoxanthin, mg/mL; the correlation coefficient R^2^ = 0.9999.

Determination of the fucoxanthin of kelp: According to Sui et al. [26], with minor modifications, the high-performance liquid chromatography (HPLC, Shimadzu-SPD-20A, Japan) was used for the determination of fucoxanthin in a C18 column (4.6 × 250 mm, 5 µm, Shimadzu, Japan) and at 40 °C. Methanol was used as the mobile phase at 0.5 mL·min^−1^, and the elution gradient was 100% methanol in 10 min. The injection volume of sample volume was 10 μL. Detection was performed by a diode array detector (DAD) at 450 nm. The corresponding peak areas of the untreated and processed kelp samples were then determined according to the method described above, and the content of fucoxanthin in the kelp extract was calculated by substituting into the standard curve. The fucoxanthin content of kelp processed by each method was measured three times.

### 2.7. Determination of Chlorophyll Content

Chlorophyll was extracted with a few modifications, as described by Havlíková et al. [27]. In summary, 1 g of raw and processed fresh *L. japonica* samples were mixed with 5 mL of acetone and 1 mL of ultrapure water in sequence and after an ultrasonic treatment for 2 h. The resulting mixture was centrifuged at 4000 rpm for 10 min. Supernatant was then evaporated (200 rpm, 40 °C) in a rotary evaporator (SY-2000, Shanghai Yarong Biochemical instrument Factory, Shanghai, China) to obtain a dry extract. Then, an equal ratio of acetone (HPLC) was used to restructure. The obtained solution was filtered through 0.45 μm nylon microporous filter membrane, and then, 1 mL was transferred to a brown sample bottle until further use.

Drawing of the standard curve of chlorophyll a: First, 1 mg of chlorophyll a was accurately weighed and dissolved with 1 mL chromatographic acetone, and the concentration of 1 mg/mL of chlorophyll a standard reserve solution was configured and stored at −20 °C in the dark. Then, a certain amount of chlorophyll a standard solution was absorbed accurately and diluted to the chlorophyll a standard working solution of 0.05, 0.1, 0.5, 1.0, 5.0, 10.0, and 50.0 μg/mL, which was placed in brown sample bottle for testing. The equation of the standard curve was obtained as y = 52,023x + 5783, where y is the peak area, and x is the concentration of chlorophyll a, μg/mL; the correlation coefficient R^2^ = 0.9979.

Determination of chlorophyll a: Determination of chlorophyll was performed by high-performance liquid chromatography (HPLC, Shimadzu-SPD-20A, Shimadzu (Shanghai) Experimental Equipment Co., LTD, Tokyo, Japan) in a C18 column (4.6 × 150 mm, 5 µm, Shimadzu, Japan) and at 30 °C. Methanol was used as the mobile phase at 0.5 mL·min^−1^, and the elution gradient was 100% methanol in 20 min. Detection was performed by a diode array detector (DAD) at 430 nm. The injection volume of sample volume was 10 μL. The corresponding peak areas of the untreated and processed kelp samples were determined according to the above method, and the content of chlorophyll a in the kelp extract was calculated by substituting into the standard curve, respectively. The chlorophyll a of kelp processed by each method was measured three times.

### 2.8. Gas Chromatography–Mass Spectrometry (GC–MS) Analysis

GC-MS analysis was performed using an Agilent 7890B instrument (Agilent Technologies, Inc., Zurich, Switzerland) equipped with an Agilen 5977B mass selective detector (MSD). The capillary column used was a VF-WAXms column, which was maintained at 35 °C for 3 min, then heated to 75 °C at 3 °C·min^−1^ for 3 min, finally heated to 240 °C at 5 °C·min^−1^, and held for 5 min. The temperature at the entrance to the gas chromatograph was 250 °C. Helium was used as the carrier gas at a flow rate of 1.68 mL·min^−1^. For the all samples, solid-phase microextraction fiber of polydimethylsiloxane/divinylbenzene/carboxen (PDMS/DVB/CAR) was used to extract 40 min at 60 °C, and the volatile compounds in the fiber were desorbed for 20 min. The mass spectrometer was operated in electron ionization mode at an energy of 70 ev. Signals were collected in full scan mode from *m*/*z* 50 to 550. The identification of volatile compounds were compared with those in the NIST2014 library. Confirmation of identification was completed by comparing the retention indices (RI) with those of an alkane standard solution (C9–C30) and the related literature. The volatile compounds of the kelp were measured three times after each processing method. In order to make the data more accurate and visually rigorous, the data measured were expressed in a relatively quantitative form.

### 2.9. Scanning Electron Microscope (SEM) Measurement

The microstructural observation of freeze-dried kelp was performed by using SEM. Images of the freeze-dried kelp surface microstructure were obtained at an accelerating voltage of 5 kV at 100× and 500× magnification. SEM images of kelp processed by each method were measured three times.

### 2.10. Statistical Analysis

All of the experiments were conducted at least in triplicate. Data are presented as the means ± standard deviation. Statistical comparisons were performed by one-way analysis of variance with Turkey’s test (*p* < 0.05). Statistical analysis was determined with SPSS 20.0 (SPSS Inc., Chicago, IL, USA) and Origin 2019 (OriginLab, Northampton, MA, USA).

## 3. Results

### 3.1. Effects of Cooking Methods on the Quality of L. japonica

#### 3.1.1. Effects of Cooking Methods on the Color of *L. japonica*

Color is one of the important plant-based food sensory parameters, as it affects consumer acceptance and recognition of plant-based foods [28]. As shown in Figure 1, the research found that the color of the *L. japonica* changed after different processing methods. Color analyses are reported in Table 1 for raw and cooked *L. japonica*. Among them, compared with raw *L. japonica*, the *L** value of *L. japonica* processed by other cooking methods had no significant change except for the significantly higher *L** of baked *L. japonica*. After baking, the *L** value of raw kelp rose from 20.15 to 27.58, indicating an increase in the *L** value of the kelp after baking. Compared with raw kelp, the *a** value of kelp changed significantly after processing. The *a** values of processed kelp changed significantly compared to raw kelp. The *a** values decreased from −0.22 to −0.45 and −0.55 for the blanched and steamed treated kelp, respectively, indicating an increase in kelp greenness values, and increased from −0.22 to 0.48 and 0.67 for the boiled and baked kelp, respectively, indicating a decrease in kelp greenness values. Previous studies also found that boiled broccoli stems and florets, green beans, and carrots took on a more intense green color [15,29]. The color of green vegetables was mainly related to the content of chlorophyll pigment in the plant material, and the increase in green color (−*a**) might be related to the change in surface reflective properties and the depth of light penetration into the cooked vegetable tissue as the cells lose air and other dissolved gases, which are replaced by cooked water and cell juice. It was also possible that chlorophyll a and b produced other green-degradation products [30].

Compared with untreated kelp, the *b** value of kelp was significantly reduced after blanching (from 5.78 to 4.45), indicating that its yellowness was weakened. The possible reason was that cooking and heating changed the content of phenolic substances in vegetables. In the presence of oxygen, the polyphenol oxidase zymogen was activated, and the phenolic substances were oxidized into quinone compounds by the catalytic action of the enzyme, and the quinone substances were polymerized to form melanin and caused browning, which in turn caused the *b** value to decrease. Another possible reason was the conversion of chlorophyll to pheophytin after blanching [31]. In general, compared with untreated *L. japonica*, the steamed-treated sample had the smallest Δ*E* and between small to medium color difference (0.5–1.0), while bleached- and boiled-treated kelp had medium color differences (1.0–2.0), and baked-treated kelp showed very large color changes (>4.0). In summary, steaming helped to reduce the color change of the kelp.

#### 3.1.2. Effects of Cooking Methods on the TPA of *L. japonica*

The effects of different processing methods on the texture of *Laminaria japonica* are shown in Table 1. The texture analyzer was used to determine the hardness, cohesiveness, adhesiveness, chewiness, and resilience of the untreated *L. japonica*. The results showed that, with the exception of adhesiveness, the textural characteristics of kelp cooked in different processing methods changed more significantly compared to raw kelp (*p* < 0.05). Among them, the texture of boiled kelp was most similar to that of raw kelp (no significant difference, *p* > 0.05) except for the resilience. On the contrary, the texture of kelp after baking changed most obviously. Hardness is a physical index that describes the strength required for chewing food. Moreover, hardness is probably the most relevant texture feature for solid foods and plays a key role in consumer acceptance and market value. Compared with raw kelp, the hardness of processed kelp was significantly reduced (*p* < 0.05). The decrease in the hardness value of the kelp was probably due to the thermal decomposition of the polysaccharides (alginate) in the kelp cell walls, which formed a porous structure, resulting in a decrease in its hardness [20]. Similar to our results, Jiang et al. found that the hardness values of *Undaria pinnatifida* decreased significantly after different cooking methods compared to the raw samples [18]. In addition, using artichokes, green beans, broccoli, and carrots as ingredients, Guillén et al. [15] studied texture changes of these ingredients under different cooking conditions. It was found that the hardness of these foods would be reduced regardless of the method, time, ingredients, and heat treatment. Similarly, Monalisa et al. [32] studied the change in the hardness of pumpkins when boiled at different lengths. The result showed the hardness of all the pumpkin decreases. It might be due to the degradation of pectin substances in pumpkin, such as galacturonic acid, to soluble pectin during the boiling process. Compared with untreated kelp, the chewiness of steamed kelp was significantly reduced (*p* < 0.05). Resilience refers to the extent to which the deformed sample returns to its original shape under the same conditions. Similarly, the resilience of kelp after all different processing methods changed except for that under blanching. Among them, the most noticeable decrease in resilience was in the baking, followed by boiling, steaming, and blanching (*p* < 0.05). In general, the different processing methods had a significant effect on the texture of the kelp, with boiling being more beneficial in maintaining the texture of the kelp, while baking was the most significant in reducing the texture of the kelp.

#### 3.1.3. L-NMR Analysis

The transverse relaxation time of L-NMR can reflect the water state and distribution. Generally, water–macromolecule interactions in the food network structure could affect the water relaxation time. As shown in Figure 2, the relaxation curves of different samples exhibited a multi-exponential behavior due to different structural elements, and the relaxation time of all three components was affected by four processing methods. This phenomenon might be related to the fact that the longer cooking time and higher temperature could promote the internal water losses [33]. The test results after inversion showed that there are 3~4 peaks, including T_2b1_, T_b22_, T_21_, and T_22_. According to the peak time, it can be considered that the four peaks, respectively, represent the three different states of water in the kelp during the heating process: bound water (T_2b_, 0–10 ms), immobilized water (T_21_, 10–100 ms), and free water (T_22_, 100–1000 ms). Among them, T_2b1_ is considered to be strongly bound water, and T_2b2_ is weakly bound water [34]. The results showed that the moisture changes in the kelp were most pronounced after baking. Except for the bound water, the immobilized water and free water in the baked kelp were almost completely lost. This could be due to the high temperature during the baking process, which prompted a large amount of water to migrate from the inside of the kelp to the outside. In addition, the water relaxation times change of steamed kelp were also different from kelp processed by other cooking methods. Both T_21_ and T_22_ shifted significantly to the left, indicating that the water inside the steamed kelp gradually changed to bound water, which may be related to the greater exposure of the kelp to the air during the steaming process. Similarly, Jiang et al. found that the water in *Undaria pinnatifida* migrated towards bound water after steaming [18]. However, the T_22_ peak area of the kelp treated by other methods became larger, indicating that the free water inside increased. The possible reason was that in the heat-treatment process, the hydrogen bonds between the water molecules in the kelp and the large molecules such as proteins are broken, so more free water could be accommodated [35]. Compared to the other processing groups, the T_22_ peak area was smaller for steaming and blanching, suggesting that they were processed under milder conditions that were conducive to retaining the moisture in the kelp.

### 3.2. Influence of Different Processing Methods on the Phloroglucinol of L. japonica

Phlorotannins are a group of major polyphenol secondary metabolites found only in brown algae and are known for their bioactivities and multiple health benefits [25,36]. At the same time, the phlorotannins are a complex molecular combination formed by the polymerization of phloroglucinol [37]. Therefore, phloroglucinol was used as the standard, and phloroglucinol content in kelp treated by different processing methods was compared. The determination results were shown in Table 2. The content of phloroglucinol in raw *L. japonica* was 3.42 mg/g, and the content of phloroglucinol in the *L. japonica* was significantly decreased after the four cooking methods (*p* < 0.05). Among them, the most significant decrease in the phloroglucinol content was observed in blanched kelp (1.98 mg/g), followed by steamed kelp (2.28 mg/g), baked kelp (2.33 mg/g), and boiled kelp (2.83 mg/g). Amorim et al. reported changes in the content of bioactive compounds in seaweed after cooking. The results showed that the content of phloroglucinol in cooked wakame decreased from 3986 mg/kg to 2600 mg/kg, but the content of phloroglucinol in fresh kelp was 1175 mg/kg and increased by 1409 mg/kg after cooking [38]. In addition, Amorim Carrilho et al. studied the changes of bioactive compounds in dried seaweed *Himanthalia elongata* under different cooking conditions and processes. They found that rehydrated seaweed had the highest content of phloroglucinols (3963.0 mg/kg), followed by boiling (2528.1 mg/kg) and steaming (1234.9 mg/kg) [39]. Overall, the different processing methods had a significant effect on the phloroglucinol content of seaweeds. All of these cooking methods are thermal processing, so the heating mode, heating temperature, and heating time affect the formation of phenolic compounds in seaweeds [40]. Some studies have reported that under the conditions in thermal processing, polyphenols might undergo isomerization and decarboxylation, thus converting to smaller molecules or more complex compounds [41,42]. In addition, heat could cause cellular damage and induce enzymatic activity and oxidative stress, leading to thermal degradation of phenolic compounds [41]. In this experiment, there was no significant difference in the phloroglucinol content of steamed kelp, boiled kelp, and baked kelp. (*p* > 0.05). Therefore, relatively speaking, boiling might be a better method to retain the content of phloroglucinol in kelp. However, to date, there has been limited research on the effects of cooking on the levels of phloroglucinol from brown algae.

### 3.3. Study on the Effect of Different Processing Methods on the Content of Fucoxanthin in L. japonica

Fucoxanthin is one of the most abundant carotenoids unique to brown algae, and its total amount accounts for about 10% of the total carotenoid production in nature [43]. In addition, fucoxanthin can also protect chlorophyll from damage caused by excessive light energy [44]. The fucoxanthin content of the *L. japonica* treated with different processing methods and the untreated kelp was measured by high-performance liquid chromatography. The results are shown in Table 2. The results show that the fucoxanthin content of *L. japonica* treated by different cooking methods was detected, and the content decreased significantly (*p* < 0.05). As higher temperatures caused a decrease in the content of fucoxanthin, this indicated an increase in the formation of epoxides in the chemical structure. In addition, it was found that high temperatures could promote increased isomerization reactions, with the total fucoxanthin content and the concentration of the all-trans isomer decreasing over time, while the total cis isomer increased [45]. Overall, oxidation and isomerization reactions were the main mechanisms for the loss of fucoxanthin when temperatures were increasing [46]. Compared with the untreated kelp (1.12 μg/g), the loss of fucoxanthin content in baked (0.37 μg/g) and blanched (0.44 μg/g) kelp was the most obvious, followed by steamed (0.69 μg/g) and boiled (0.68 μg/g) kelp. This result was consistent with the previous *b** results; i.e., steaming and boiling were more favorable to retaining the fucoxanthin in the kelp, thus making them contribute more to the kelp *b** and have higher *b** values. Danowska-Oziewicz et al. [11] also found a decrease in carotenoid content in steamed broccoli and boiled asparagus. In the experiment of Zhang and Hamauzu [47], the content of carotenoids declined gradually during conventional boiling in water in a similar manner. The degradation of carotenoids as a result of thermal treatment of broccoli was also reported by Guillén et al. [15]. One possible reason for the variation in carotenoid content in cooked vegetables was oxidation due to the presence of double bonds in their structure and their high dependence on oxygen and light access and high temperatures during cooking [48]. Based on the experimental results, steaming and boiling appeared to be beneficial in reducing the loss of fucoxanthin content of *L. japonica*.

### 3.4. Effects of Different Processing Methods on the Chlorophyll Content of L. japonica

During the processing of *L. japonica*, special attention should be paid to the fact that their main pigment, chlorophyll, is easily decomposed into derivatives, making it less attractive to the consumer, so chlorophyll is also a key indicator for evaluating different processing methods. Chlorophyll is a natural yellow-green pigment and the most widely used pigment in nature. It can convert light energy into biological energy by plants, which is easy to obtain from plants and algae. The chlorophyll a content of *L. japonica* treated with different processing methods and untreated was determined by high-performance liquid chromatography. The results showed that the chlorophyll a was not detected in the *L. japonica* of this variety; i.e., no value contributed to the *a** values in the kelp, so the authors speculated that the chlorophyll in the *L. japonica* was more likely to be present in other chlorophyll forms and their derivatives. Because chlorophyll is very unstable, it breaks down when exposed to light, acids, bases, oxygen, and oxidizing agents, resulting in the production of various chlorophyll derivatives [49]. Chlorophyll molecules are subjected to microwave heating or high-temperature steam conditions, where the ester group is removed to produce pyrochlorophyll. In addition, the α-H site of the chlorophyll molecule is susceptible to substitution by hydroxyl groups, resulting in derivatives of hydroxychlorophylls. Chlorophyll a derivatives are susceptible to isomerization under heat treatment, forming the isomer chlorophyll a′, and chlorophyll a′ reaches a maximum after microwave, hot blanching, baking, and steam treatment and is relatively easier to produce than chlorophyll a′ [50]. As a result, active sites in the chlorophyll molecule, such as ester bonds, magnesium atoms, and the alpha-H binding site of the carbonyl group, could easily be lost or broken during the heat treatment, resulting in the formation of a wide range of chlorophyll derivatives [51]. Similar to our results, Benlloch-Tinoco et al. studied the effects of microwaving, thermal processing, and storage on kiwifruit pigments and found that conventional heating degraded chlorophyll more readily than microwave treatment and that no chlorophyll b was detected in kiwifruit puree treated with conventional heating, possibly due to more rapid degradation in the presence of chlorophyllase or other enzymes [52].

### 3.5. Volatile Compounds

Table 3 lists the volatile compounds identified and quantified in raw, blanched, steamed, boiled, and baked *L. japonica*. The results were the arithmetic mean ± standard deviation of three samples. The results show that eight flavor substances were detected in raw *L. japonica*. Among them, the content of 1-octen-3-one was the most abundant, reaching 48.32%, followed by the content of 1-methoxy-2-propyl acetate (21.33%) and 1,4-di-iso-propylnaphthalene (21.33%). However, only four flavors were detected in the kelp after the blanched treatment, and compared with the untreated kelp, all of these four flavors were newly generated. Among them, the content of trans-β-ionone and benzaldehyde was the most abundant, reaching 37.80% and 35.06%. It is worth noting that trans-β-ionone is an important carotenoid derivative [53] that can be produced by thermal degradation or by enzymatic reactions [54], so it could be speculated that carotenoids (possibly fucoxanthin) from kelp were produced by degradation during blanching processing because blanching did significantly reduce the fucoxanthin content in kelp (from 1.12 to 0.44), as can be seen in Table 2. The steamed kelp had more flavor substances, the same as the raw kelp, both with 1-methoxy-2-propyl acetate and 1-octen-3-one flavor substances. In addition, nine flavor substances were newly formed after steaming. Of the nine newly generated flavor substances, two were identical to those found in blanched kelp, namely benzaldehyde and acetophenone. Benzaldehyde was the most abundant flavor compound in steamed kelp (24.23%), similar to the blanched kelp. Benzaldehyde could be produced by derivation from amino acids [55] or by oxidation of trans-cinnamic acid [56].There were seven flavor substances in the boiled kelp, among which 1-methoxy-2-propyl acetate (66.21%) and 2-heptenal (Z)-(7.91%) were the same as that in the raw kelp, but it was the same as that in the steamed kelp, where both produced styrene. The 1-Methoxy-2-propyl acetate content increased from 21.33% s to 66.21%, indicating that cooking significantly increased the ester content of kelp. This might be due to the esters generated by decarboxylation and substitution of lipids in kelp during the cooking process [57]. Compared with raw kelp, baked kelp showed the most significant changes in flavor compounds. A total of 30 flavor compounds were detected in the baked *L. japonica*, among which 4 were the same as those in the raw kelp. In addition, unlike the raw kelp, the naphthalene 1,2-dihydro-1,1,6-trimethyl- was the same as the blanched kelp, and furan, 2-pentyl-, styrene, and the phenol 3,5-bis(1,1-dimethylethyl)- were the same as the steamed kelp. Styrene and 2’,3’,4’trimethoxyacetophenone in boiled kelp were also detected in baked kelp but at lower levels. Furthermore, the content of decanal and pentadecane were the most abundant at 21.87% and 21.86%, respectively. Some identified compounds were also found in previous works such as that of Pina et al. [58] and Mirzayeva et al. [59]. In general, all cooking methods induced an increase in the content of aldehydes compared to raw kelp. A similar phenomenon was found by Garrido-Bañuelos et al. [60]; i.e., longer heating times induced a significant release of several aldehydes from seaweed (*Laminaria digitata*) suspensions. This might be due to the oxidation of fatty acids [61] in the kelp due to heating and the formation of unsaturated aldehydes (e.g., benzaldehyde in blanching and benzaldehyde and (E)-2-heptenal in steaming) and saturated aldehydes (decanal in baking), all of which were associated with a “fatty” odor [62]. In addition, the concentration of aldehydes in some processed foods might increase after heat processing due to chemical reactions that occur under high-temperature heating conditions, such as degradation of amino acids or carbohydrates [63]. In conclusion, steaming and boiling were better for maintaining the original flavor substances of the kelp, while baking was more beneficial for producing new flavor compounds.

### 3.6. SEM Analysis

Using scanning electron microscope to observe the kelp after different processing methods (Figure 3), it could be seen that there was no obvious change in the microstructure of the blanched *L. japonica* compared with the raw *L. japonica*. Compared to raw kelp, the steaming-treated kelp became thinner overall, with larger pore sizes, distinct cavities, and a significantly reduced portion of the network structure. Compared with the control group, the structure of the boiling-treated kelp changed significantly, with a marked increase in network structure and greater density. Similar to steamed kelp, there were cavities in the kelp after boiling. The possible reason was that the jelly formed in the kelp after cooking was dehydrated after freeze drying. Among the kelp processed by various processing methods, the most obvious change was the kelp after baking. The kelp became very thin, and the porous structure disappeared completely, but it became a dense, scaly structure. The probable reason for this was the fact that the water in the *L. japonica* was almost completely lost during the baking process. There are many other studies that confirmed that the structure of food changes after cooking or heat treatment. For example, Yang et al. [64] observed that the surface of the fried potato sticks changed under heating. In summary, blanching and steaming were found to be favorable for maintaining the original microstructure of the kelp, while baking significantly altered the porous reticulation of the kelp.

## 4. Conclusions

The results showed that all four processing methods affected the texture and color of the kelp to varying degrees, with steaming showing the least change in color, boiling the least change in texture, and baking the most significant change in color and structure. In addition, all processing methods significantly reduced the content of phloroglucinol and fucoxanthin in kelp (*p* < 0.05); however, both steaming and boiling were effective in suppressing the reduction in the content of these two active substances. In addition, blanching and boiling treatments reduced the flavors of kelp, while steaming and baking enhanced the flavor of kelp. In conclusion, steaming and boiling were the optimal processing methods to maintain the quality of kelp processing and reduce the loss of active substances from the kelp. Overall, this research provides a scientific basis for promoting the intensive processing of *L. japonica*, improving the quality and economic benefits of *L. japonica* processing and transformation, and providing a certain scientific basis for the implementation of reasonable processing of *L. japonica*.

## Figures and Tables

**Figure 1 foods-12-01619-f001:**
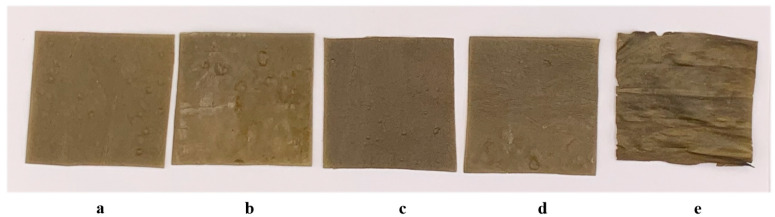
Raw and cooked *L. japonica* by different methods. The processing methods of *L. japonica* from left to right are (**a**) raw; (**b**) blanching; (**c**) steaming; (**d**) boiling; (**e**) baking.

**Figure 2 foods-12-01619-f002:**
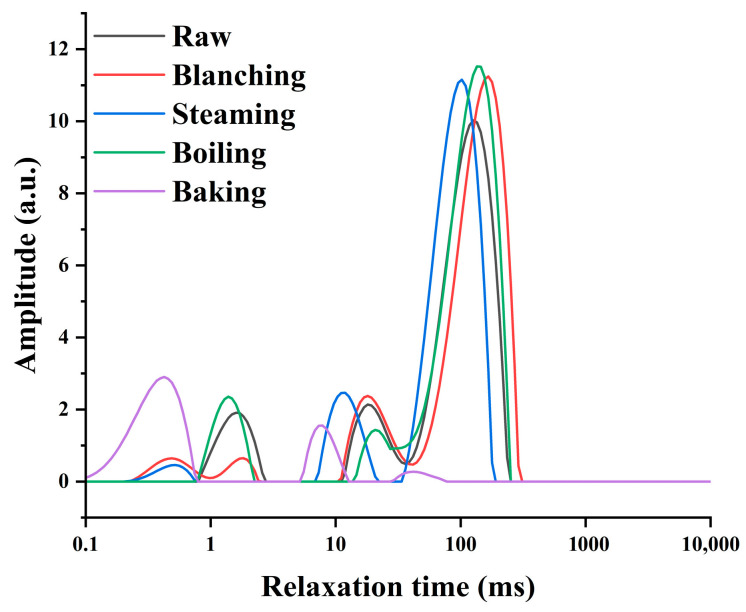
Distribution of T_2_ relaxation times in *L. japonica* cooked with four different cooking methods.

**Figure 3 foods-12-01619-f003:**
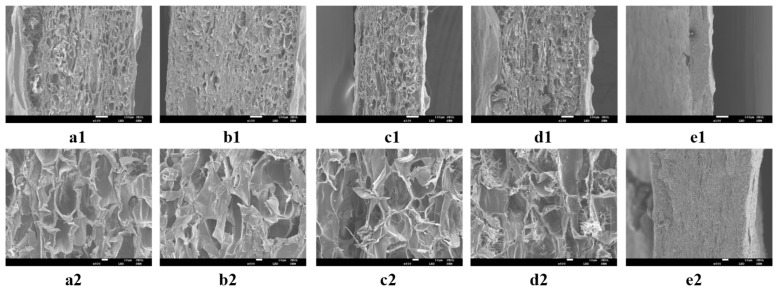
The microstructure of *L. japonica* treated by different processing methods was observed by scanning electron microscope.×100: (**a1**) raw; (**b1**) blanched; (**c1**) steamed; (**d1**) boiled; (**e1**) baked. ×500: (**a2**) raw; (**b2**) blanched; (**c2**) steamed; (**d2**) boiled; (**e2**) baked.

**Table 1 foods-12-01619-t001:** Influence of different processing methods on the color and TPA of *L. japonica*.

		Raw	Blanched	Steamed	Boiled	Baked
Color	*L**	20.15 ± 1.15 ^b^	21.17 ± 0.76 ^b^	19.96 ± 1.78 ^b^	20.95 ± 0.84 ^b^	27.58 ± 1.40 ^a^
	*a**	−0.22 ± 0.11 ^b^	−0.45 ± 0.14 ^c^	−0.55 ± 0.18 ^c^	0.48 ± 0.19 ^a^	0.67 ± 0.22 ^a^
	*b**	5.78 ± 0.83 ^a^	4.45 ± 0.40 ^b^	5.29 ± 0.82 ^ab^	5.30 ± 0.96 ^ab^	4.97 ± 0.98 ^ab^
	Δ*E*	nd	1.69	0.61	1.18	7.52
TPA	Hardness (g)	1513.73 ± 103.87 ^a^	1348.21 ± 118.54 ^b^	1085.80 ± 110.07 ^c^	1495.20 ± 105.38 ^a^	360.05 ± 112.053 ^d^
	Cohesiveness	1.04 ± 0.17 ^a^	0.99 ± 0.02 ^ab^	0.96 ± 0.02 ^ab^	0.99 ± 0.03 ^ab^	0.89 ± 0.14 ^b^
	Adhesiveness (g.s)	0.67 ± 0.02	0.67 ± 0.04	0.60 ± 0.04	0.65 ± 0.01	0.58 ± 0.31
	Chewiness (g)	1044.65 ± 160.11 ^a^	976.93 ± 304.12 ^a^	626.84 ± 105.97 ^b^	953.46 ± 57.72 ^a^	140.44 ± 126.20 ^c^
	Resilience	0.53 ± 0.03 ^a^	0.48 ± 0.05 ^ab^	0.42 ± 0.05 ^bc^	0.39 ± 0.02 ^c^	0.30 ± 0.08 ^d^

Data are expressed as mean ± standard deviation (*n* = 9). Means in columns followed by different letters differ significantly (*p* ≤ 0.05). nd, not determined.

**Table 2 foods-12-01619-t002:** Effects of different processing methods on the contents of polyphenols, fucoxanthin, and chlorophyll in *L. japonica*.

	Phloroglucinol mg/g	Fucoxanthin μg/g	Chlorophyll a μg/g
Raw	3.42 ± 0.13 ^a^	1.12 ± 0.14 ^a^	-
Blanched	1.98 ± 0.05 ^c^	0.44 ± 0.01 ^c^	-
Steamed	2.28 ± 0.16 ^bc^	0.69 ± 0.03 ^b^	-
Boiled	2.83 ± 0.43 ^b^	0.68 ± 0.02 ^b^	-
Baked	2.33 ± 0.09 ^bc^	0.37 ± 0.01 ^c^	-

Data are expressed as mean ± standard deviation (*n* = 3). Means in columns followed by different letters differ significantly (*p* ≤ 0.05). -, not detected.

**Table 3 foods-12-01619-t003:** Flavor compounds produced by different processing methods of *L. japonica*.

	Raw (%)	Blanched (%)	Steamed (%)	Boiled (%)	Baked (%)
1-Methoxy-2-propyl acetate	21.33 ± 10.03	-	14.46 ± 4.21	66.21 ± 8.95	-
1-Octen-3-one	48.32 ± 16.26	-	17.17 ± 6.76	-	3.02 ± 0.11
2-Heptenal, (Z)-	2.77 ± 2.71	-	-	7.91 ± 2.48	1.34 ± 0.13
1-Pentanol, 3-methyl-	0.96 ± 0.90	-	-	-	-
Nonanal	0.73 ± 0.12	-	-	-	1.70 ± 0.28
2-Octenal, (E)-	1.50 ± 2.30	-	-	-	1.66 ± 0.16
Propanoic acid,2-methyl-, 1,1′-(2-ethyl-1-propyl-1,3-propanediyl) ester;	9.92 ± 6.82	-	-	-	-
2(4H)-Benzofuranone, 5,6,7,7a-tetrahydro-4,4,7a-trimethyl-	2.58 ± 0.66	-	-	-	-
1,4-Diisopropylnaphthalene	21.33 ± 10.03	-	-	-	-
Benzaldehyde	-	35.06 ± 0.39	24.23 ± 3.95	-	-
Acetophenone	-	24.53 ± 2.24	3.07 ± 0.31	-	-
Naphthalene,1,2-dihydro-1,1,6-trimethyl-	-	2.61 ± 0.71		-	0.96 ± 0.63
trans-β-ionone	-	37.80 ± 1.95		-	-
Furan, 2-pentyl-	-	-	14.83 ± 3.65	-	8.88 ± 0.34
Styrene	-	-	2.44 ± 0.87	3.20 ± 1.67	1.13 ± 0.20
2-Heptenal, (E)-	-	-	11.54 ± 1.83	-	-
5-Ethylcyclopent-1-enecarboxaldehyde	-	-	1.59 ± 0.07	-	-
Benzaldehyde	-	-	8.86 ± 0.87	-	-
1-Octanol	-	-	1.28 ± 0.08	-	-
Phenol, 3,5-bis(1,1-dimethylethyl)-	-	-	0.53 ± 0.14	-	0.13 ± 0.03
3-Octanone, 2-methyl-	-	-	-	6.27 ± 2.36	-
2′,3′,4′ Trimethoxyacetophenone	-	-	-	1.04 ± 0.47	0.04 ± 0.00
2,4-Di-tert-butylphenol	-	-	-	11.54 ± 5.06	-
Dibutyl phthalate	-	-	-	3.84 ± 2.87	-
2,3-Pentanedione	-	-	-	-	1.70 ± 0.08
2-Heptanone	-	-	-	-	0.87 ± 0.08
2-Hexenal, (E)-	-	-	-	-	0.37 ± 0.00
1,3,5,7-Cyclooctatetraene	-	-	-	-	0.64 ± 0.05
2,3-Octanedione	-	-	-	-	7.23 ± 0.21
1-Tridecene	-	-	-	-	0.11 ± 0.01
1-Octen-3-ol	-	-	-	-	10.03 ± 0.41
2-Dodecyne	-	-	-	-	0.89 ± 0.07
Decanal	-	-	-	-	21.87 ± 1.10
Pentadecane	-	-	-	-	21.86 ± 1.08
3,5-Octadien-2-one	-	-	-	-	1.92 ± 0.19
2-Furancarboxaldehyde, 5-methyl-	-	-	-	-	2.29 ± 0.35
2-Octen-1-ol, (E)-	-	-	-	-	0.97 ± 0.07
Dodecanal	-	-	-	-	0.77 ± 0.53
1-Cyclohexene-1-carboxaldehyde, 4-(1-methylethyl)-	-	-	-	-	1.62 ± 0.94
2,4-Decadienal, (E,E)-	-	-	-	-	2.87 ± 0.02
Hexanoic acid	-	-	-	-	3.61 ± 0.04
3-Buten-2-one, 4-(2,2,6-trimethyl-7-oxabicyclo[4.1.0]hept-1-yl)-	-	-	-	-	0.15 ± 0.00
Methyl tetradecanoate	-	-	-	-	0.49 ± 0.04
1H-Pyrrole-2,5-dione, 3-ethyl-4-methyl-	-	-	-	-	0.79 ± 0.13
Diethyl Phthalate	-	-	-	-	0.07 ± 0.00

The data are expressed in a relatively quantitative form and presented as mean ± SD (*n* = 3). -, not detected.

## Data Availability

Data is contained within the article.

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
