# Peer review of "Effects of Different Processing Methods on the Quality and Physicochemical Characteristics of Laminaria japonica"

_foods, 2023, doi:10.3390/foods12081619_

Round 1

Reviewer 1 Report

This study aims to select the best way of cooking to maximize the retention of active substances and to improve the processing characteristics of the product.

The originality and quality of this work are indisputable. therefore, this reviewer makes small suggestions to improve its quality.

Introduction

The introduction of the work, as well as the discussion, are based on information about other plants and, rarely, on algae of the species L. japonica, or even other algae. This fact suggests the scarcity of information about these.

Because of this, I propose that the authors present in the introduction what is state-of-the-art stability studies with other varieties of algae.

The authors mention that research on the difference in processing characteristics and bioactive components of L. japonica under different cooking methods is still incomplete (Lines 88-90). In this way, I question whether studies of this nature are non-existent or insufficient? Being insufficient, I request that the works and underlying basic variables be cited. In the end, it should very clearly delimit the gaps not filled by the existing studies that one wants to fill with the present study.

Methods

Why were these treatments selected? What parameters were used for the selection of these treatments? Are these treatments used by the population for the consumption of this seaweed? What parameters were used to define the conditions of each treatment? The presence of this information is essential since it is ideal that they are similar to those used during the actual preparation by the population

Discussion

Throughout the discussion, the authors need to do a deep citations check. At points such as Line 242, the authors mention that “Previous studies have also found...” but substantiate this information with only one reference. Furthermore, there are passages like the one present in lines 355-377 that notably lack reference.

Conclusion

The conclusion of the work is supported by the results. However, it is not fully aligned with the objectives of the work. The work aimed to select the best way of cooking method to maximize the retention of active substances and to improve the processing characteristics of the product. However, the conclusion brings different considerations about each method, without categorically explaining which is the best.

Reviewer 2 Report

The study of food processing by traditional and alternative methods and its effect on properties has been studied for a long time, however, the focus on the changes that occur in bioactive compounds due to processing are more novel and current studies, and specifically there has been a focus on products such as kelp.

But some changes should be made to improve the description of the study, as well as the presentation of results.

Comments:

Introduction

Page 2, line 92: Why the treatments are named “domestic cooking methods” instead of traditional processing methods? With that name it seems that the information or results obtained are only applicable to the kitchen or direct to the consumer of the kelp. But I think this could have other applications, right?

A paragraph could be added in the introduction regarding the contribution, impact or technical applications of the information obtained.

Materials and methods:

Indicate the number of measurements for each method, not indicated for example in TPA and moisture methods.

Section 2.5, indicate “Determination of polyphenolic compounds content”. What polyphenols were determined? could, the authors, specify the standards used and the concentration of the calibration curve, to improve the description of the method.

Section 2.6: Could, the authors, specify the concentration of the calibration curve, to improve the description of the method.

Section 2.7: Could, the authors, specify the concentration of the calibration curve, to improve the description of the method; and take the opportunity to indicate that only chlorophyll a was used as the standard.

Results and discussions:

Section 3.1.1. describe the color trends with the values obtained for the parameters: L, a and b; for example: b presented values from 4.45 to 5.78, indicating the tendency to yellow color.

Table 1.

Table foot: Not only color data are presented in the table, then "values are expressed in colorimetric units" is not adequate.

There are some problems in the alignment of the rows, is it possible to present the hardness in kg units, to reduce the numbers presented?

Some data shown a high standard deviation, is the data correct?

Table 2.

Check the table foot, is correct?

Fit title to table size

Means comparison are presented by column, not by row, correct?

Why present the results of Chlorophyll a in table?

Section 3.2: The authors could add the reasons why there was a change in the Phloroglucinol content due to the type of processing (based on literature or reported data), to deepen the discussion and not just a description of results.

Page 9, lines 385-387: The authors present a speculation about Chlorophyll a, based on what? It may be better not to present the “chlorophyll a” results, as this is speculation only. It can only be described that “chlorophyll a” was not detected.

Section 3.3: A degradation of fucoxanthin is written, information related to the degradation products of the compound due to the action of temperature (supported by the literature or reported data) could be added to deepen the discussion of the results.

Table 3:

In the first line the percentage unit (%) is presented, then the unit of the numbers within the table can be eliminated.

Why are the results in %? In the footer of the table, it is indicated that they are relative numbers but the authors could describe better this in section 2.8 of materials and methods.

Reviewer 3 Report

General Comments:

The manuscript presents a promising scientific article; however, it requires significant revisions and corrections. Certain parameters require proper referencing for their alterations, and the assessment of volatile components must be conducted thoroughly. This paper necessitates validation and rewriting.

Specific Comments:

Line 2: The title can be changed as it contains too many conjunctions. "Quality and physical and chemical properties" can be changed to "quality and physicochemical characteristics"

Line 11: "blancing" should be "blanching"

Line 16: "hardness and chewiness was" should be "hardness and chewiness were"

Line 45: "L japonica" - Should be full-stop after L

Line 51: "show" should be "shows"

Line 51: There should be a space after the reference [10]

Line 63: "Worst sensory" - It should be worst sensory quality or worst sensory perception.

Line 69-70: "have caused the content of phenolic components change" can be changed to "have caused changes in phenolic content of components.

Line 75: "high temperatur" should be "high temperature"

Line 84: HPHT temperature starts from 150°C. Kindly verify this.

Line 103: Full stop after Dr. Mao

Line 108: Preparation of sample - This section is written in instruction form. It should be changed to past tense procedures that were actually performed in laboratory

Line 109: Laminaria japonica should be italicized

Line 116: "blancing" should be "blanching"

Line 122: "Boild" should be "boiled"

Line 127: "contact" should be "cooled"

Line 149: British should be changed to proper city in UK

Line 163: "Collected the supernatant" should be "Supernatant was collected"

Line 178: Remove "briefly"

Line 186: "using" should be "used”

Line 197-198: This sentence should be rewritten with past tense

Line 207: "instrumen" should be "instrument"

Line 265: Table 1. Mention units for Cohesiveness, Adhensiveness, Chewiness and Reslience. Also, this table should be positioned after Section 3.1.2. Effects of cooking methods on the TPA of L. japonica

Line 268: Remove extra dot

Line 313: "he moisture" should be "the moisture"

Line 363: "blanced" should be "blanched"

Line 374: Space between "cooking [35]"

Line 384: "were not detected" should be "was not detected"

Line 398-399: "1-octene-3-one" should be "1-octen-3-one"

Line 416: Table 3 - No need to add percentage (%) symbol after every value. Symbols on the Table's first row after the labels are enough.

Table 3. One thing needs to be clarified that there are components which are present in processed (heated) kelps but not present in raw kelp. A proper literature and reason for that should be provided as to how these new compounds are formed in heated kelps. Are these through polymerization or other chemical reaction. Kindly clarify. Also, some of the compound names are written in last name, first name basis. These should be corrected in linear order.

Line 449: "blanched and boiled treatments" should be "blanching and boiling treatments"

Round 2

Reviewer 2 Report

Thank you for answering the questions and taking into account the suggestions and comments.

The description of some analyzes was improved, adding the information on the standard used and the concentration of the calibration curves for content determination, as suggested in the previous review. However, some methods were described in too much detail, so some lines can now be removed:

In section 2.5 polyphenolic compounds content: lines 200 - 204 and lines 209 - 211, are not necessary.

In section 2.6 fucoxanthin content: lines 232-238 are not necessary.

In section 2.7 chlorophyll content: lines 261-268 are not necessary.
